# In-Depth Development of a Versatile Rumen Bolus Sensor for Dairy Cattle

**DOI:** 10.3390/s24216976

**Published:** 2024-10-30

**Authors:** Gergely Vakulya, Éva Hajnal, Péter Udvardy, Gyula Simon

**Affiliations:** Alba Regia Faculty, Obuda University, 8000 Székesfehérvár, Hungary; vakulya.gergely@amk.uni-obuda.hu (G.V.); udvardy.peter@amk.uni-obuda.hu (P.U.); simon.gyula@amk.uni-obuda.hu (G.S.)

**Keywords:** precision livestock farming, digital data collection, LoRaWAN communication, intraruminal device, accelerometer, heart rate estimation

## Abstract

Precision agriculture and the increasing automation efforts in animal husbandry requires continuous and complex monitoring of the animals. Rumen bolus sensors, which are cutting-edge pieces of technology and a rapidly developing research field, present an exceptional opportunity for monitoring the health status, physiological parameters, and estrus of the animals. The objective of this paper is to provide a comprehensive overview of the development process of a new sensor development. We address the issues of conceptual design, an overview of applicable sensor modalities, mechanical design, power supply design, applicable hardware solutions, applicable communication solutions and finally the sensor detection algorithms proved in field tests. In conclusion, we present a summary of the current opportunities in the field and provide an analysis of the foreseeable trends.

## 1. Introduction

The Agriculture 5.0 trend, which is often called Precision Livestock Farming (PLF) in the livestock sector, has been gaining ground after the Industry X.0 trends since the 2000s [1]. The emergence of this trend has been triggered by a number of factors, including the optimization of the amount of labor invested and, more generally, the emphasis on automation for economic reasons, with the aim of maintaining or improving product quality [2,3,4]. The importance of sustainability has grown, and animal welfare and environmental issues play a significant role. It is important to use environmental resources responsibly, preferably through a circular economic model [5,6,7,8,9]. The traditional protocol for automation involves a cycle of measurement, evaluation, and action. Sensor systems and robotic solutions are becoming more and more common in animal husbandry [7,10,11,12,13,14,15]. The growing significance of integrated data collection, databases, and data processing needs to be mentioned. Rumen bolus sensors are becoming more important in the dairy cattle sector, which is the topic of our current article [16,17,18]. The bolus technology in the beginning phase was evaluated in detail. Experiments were conducted with bolus technology on a relevant number of animals. The retention of the bolus was investigated depending on the size, and specific gravity of the bolus, and the age and gender of the animals. However, it is not possible to obtain direct information on the possible discomfort of wearing a bolus, but based on theoretical considerations, both by measurement, it is rightly assumed that there is no discomfort effect. During the experiments, there was no detected difference between bolus-containing animals and not-containing animals in physiological parameters and behavior [19]. The conclusion can be stated that the bolus is a noninvasive technology because it can be swallowed easily without any veterinary action, it can stay in the animal for a long time, they do not get lost or fall off, and the animal will not feel any discomfort when wearing them [8,19,20].

The initial rumen boluses were developed for veterinary applications, wherein they facilitate the controlled release of pharmaceuticals or microelements into the animal’s digestive tract. Subsequently, the functionality underwent a modification, and boluses were employed for the purposes of animal identification and the actual collection of sensory data. The utilization of boluses as sensors is corroborated by the stability of the sensor within the rumen, where it persists for an extended period, potentially throughout the animal’s lifespan. This contrasts with sensors employed externally, which may be subject to degradation or loss over time [5,21]. The use of boluses as sensors is supported by the fact that the sensor is not removable from the rumen and remains in the rumen for a long time, practically for the lifetime [5,22] of the animal, unlike sensors used externally.

The application of this technique is hindered by the manner in which information is conveyed from the rumen. The initial sensors were predominantly designed for temporary implementation, with applications in pH measurement and local data collection in fistulated cows [23,24,25,26]. Subsequently, the range of modalities was expanded to encompass devices for measuring temperature [22,27,28,29,30,31,32,33,34,35] and 3D acceleration [36] data, and the corresponding radio communication technology [8,11,27,37,38] was also developed.

The field of rumen bolus technology has witnessed significant advancement in recent years, largely due to the introduction of sophisticated bolus sensors with multiple sensor modalities and extended lifespans. These advancements have enabled the monitoring and data recording throughout the entire lifespan of the animal subjects [22,33]. The rumen bolus sensor technology supports sustainable animal husbandry by continuously monitoring animals, reducing the need for human resources for animal husbandry, and supporting the farmer in animal husbandry decisions. There are many products on the market that are highly variable in quality. The use of bolus sensors on farms is becoming increasingly prevalent [16,17,18]; however, the technology has not yet achieved widespread acceptance. Firstly, the reliability of these devices has yet to be established, and secondly, the range of services they provide is not yet proven to be superior to those of other sensors [39].

Based on our experience, we can conclude that this market segment has significant potential for both commercial and scientific growth and development. Therefore, further development of these devices is required, including the extension of sensor modalities [40] and an improvement in measurement precision. Furthermore, it is imperative to conduct scientific studies to substantiate the efficacy and advantages of these devices, providing farmers with a scientific rationale for their implementation. The ultimate objective is to conduct a comprehensive evaluation of the animals and their housing conditions in farms by continuously monitoring them with sensors, using big data solutions, artificial intelligence applications, and automation solutions [1,7,41]. Our research aimed to develop a lifelong rumen bolus sensor system, which is capable of measuring many physiological attributes, including temperature, drinking, motion activity, rumination, and heart rate, instead of using multiple sensors. The system is able to process these data with the help of Artificial Intelligence (AI) on the server side to produce a complex monitoring of animal welfare, health, and farming parameters. The process of creating such a system involves multiple steps, beginning with the creation of hardware, followed by primary measurements, and finally high-level data processing. The heart rate measurement is a completely new function in rumen bolus sensors. This paper shows the development of the system, including primary data processing. However, building such a tool is a complex task and an interesting engineering challenge in the fields of electronic design, resource management, telecommunication solutions, data analysis algorithms, and artificial intelligence solutions. The objective of this article is to disseminate the scientific experience of developing such a sensor to the wider academic community, as it may prove to be of significant interest. A further important scientific topic is the validation of the data obtained from such a sensor; the significance of this subject merits a dedicated article [42,43].

In this paper, we describe the implementation of such a complex sensor development. In Section 2 the design of the system is introduced, including the global concept, mechanical design, power supply design, sensor modalities used, and communication techniques. Section 3 deals with the algorithms designed for the device, followed by a discussion and conclusions in Section 4. As the main focus of this paper is to describe the technical aspects of the bolus development, and the testing is a very large and complex process that could transcend this publication, it is only roughly described in the Section 3. Continuous testing has been going on since August 2022. The test was conducted on adult animals, including their reproductive periods. The animals that participated in the experiments were kept with the control animals in the same barn with natural conditions. The temperature and humidity were continuously measured during the experiments. The sensor validation involved comparing the measured data to the original physiological characteristics whenever possible. The validation was primarily carried out using videos recorded by a camera system and Heart Rate (HR) measurements taken with Polar HR sensors.

## 2. System Design

### 2.1. Requirements

The development team consists of veterinarians and university agricultural specialists, as well as agricultural experts from the test site. These stakeholders were involved in the preparation of the specification. Our methodological choices were based on prior expert experience, and we opted for methodologies and technologies that have already been proven to work in research tasks thus far. The problem scope was defined based on the experience of an expert team (veterinarian, farmer, agronomical expert, and engineers). The system’s objectives can be classified into three principal categories: health monitoring, illness recognition, and oestrus prediction [44].

Health monitoring is the cornerstone of any animal monitoring system. By continuously tracking key indicators such as body temperature and activity levels, these systems provide real-time data that can be analyzed to assess the overall health status of the animals. This constant surveillance ensures that any deviations from the norm are promptly detected, allowing for early intervention [30]. Essentially, health monitoring serves as an “OK” sign for the farmer, indicating that all parameters are within the required range and that no extraordinary events are occurring.

Recognizing illness at an early stage is critical for minimizing the impact of diseases on animal welfare and farm productivity. Early detection allows for timely interventions, reducing the severity and spread of illnesses. For instance, a sudden drop in activity levels or an abnormal increase in body temperature can serve as early indicators of health issues [35]. For the farmer, this functionality serves as an early warning system, signaling potential problems before they escalate into serious conditions [3]. By providing these early alerts, the system helps to improve the longevity and quality of life of the animals, while also maintaining farm productivity and minimizing economic losses associated with disease outbreaks.

Predicting oestrus is a crucial aspect where the system can generate additional profit and significantly reduce the environmental footprint [10]. If insemination does not occur during the short period of oestrus, an entire reproductive cycle is lost, leading to substantial economic losses. Furthermore, accurate prediction of natural oestrus can eliminate the need for artificial hormone programs, which are often used to induce oestrus. By relying on precise oestrus detection, farmers can enhance reproductive efficiency and reduce the use of hormones, leading to a more natural and environmentally friendly organic approach to livestock management. This not only optimizes breeding schedules and increases productivity but also promotes sustainable farming practices [2].

There are several practical products and experimental devices available for animal monitoring, which can be attached to various parts of the animals’ bodies or even implanted. Devices placed externally on the animals are often susceptible to being lost or damaged due to the animals’ movements and interactions [44]. On the other hand, implants, while secure, require complex procedures for insertion and removal, demanding special equipment and skilled personnel. In contrast, a bolus, deployed inside the rumen, offers a reliable and durable solution. By design, a rumen bolus is securely retained within the animal’s digestive system, ensuring continuous operation throughout the animal’s lifespan without the risk of loss or external damage. This method provides a robust and low-maintenance alternative for long-term health and activity monitoring [22].

The only viable method to retrieve the collected data from these devices is through a low-power radio link. However, transmitting all measurement data in real-time is impractical due to energy constraints and bandwidth limitations. Therefore, it is essential to employ data reduction techniques inside the bolus such as compression, filtering, feature extraction, or other forms of preprocessing. These methods effectively minimize the volume of data that needs to be transmitted, ensuring that the system can function over extended periods [8].

On the other end of the radio link, a receiver unit is required to relay the data packets to higher-level systems for further processing and storage. At this point, employing Low Power Wide Area Network (LPWAN) technology becomes advantageous. Two of the most promising LPWAN solutions are Narrowband IoT (NB-IoT) and Long Range Wide Area Network (LoRaWAN). NB-IoT [45] is a service provided as part of the existing cellular network, which implies that the positioning of the base stations is beyond the user’s control. Given that the signal is likely to be heavily attenuated by the bodies of the animals, this lack of control can negatively impact communication reliability. In contrast, LoRaWAN [46] allows for complete control over the network infrastructure. Gateways can be strategically installed in close proximity to the animals, ensuring robust and reliable communication. This flexibility in network deployment makes LoRaWAN a more suitable option for maintaining consistent data transmission in animal monitoring systems, especially in challenging environments where signal attenuation is a concern.

To design efficient preprocessing algorithms that run on the microcontroller, it is essential to first collect raw, unprocessed data for offline analysis. This requires the creation of a prototype bolus equipped with the same sensors as the final version but potentially utilizing a different radio chip. This prototype bolus should be capable of transmitting all sampled data continuously for a period of at least 8–10 days. Additionally, it is crucial that the inertial behavior of this prototype closely mimics that of the final version to ensure the accuracy and relevance of the data collected.

The preprocessed data are collected by the higher levels of the system, where complex decision-making processes are executed. Advanced AI and machine learning (ML) processing algorithms are employed at this stage, necessitating substantial training and external support data to ensure accurate and reliable performance. The backend infrastructure must include a robust database to store all historical data, enabling comprehensive long-term data analysis and trend identification.

Additionally, the system must feature a user-friendly frontend interface, allowing users to easily view critical data and insights. This interface should include an alert system to notify users of events that require immediate intervention, ensuring prompt responses to potential issues.

### 2.2. System Architecture

The proposed system architecture is illustrated in Figure 1. In this architecture, boluses are placed into the rumen of the cows, providing continuous monitoring of various physiological parameters. Each barn is equipped with its own LoRaWAN gateway to counteract the high attenuation caused by the animals’ bodies. It is important to note that cows situated between the bolus antenna and the gateway antenna contribute to the overall path loss, necessitating strategic placement of gateways to ensure reliable communication. In the experiment, the gateway was installed in the center of the barn, to a height of approx. 5 m. According to the Received Signal Strength Indication (RSSI) values of the data packets this resulted in a reliable connection. For long-term use with more barns, the necessity for new gateways can be determined based on the RSSI of packets from boluses of cattle being in other barns. The base area of each barn is approx. 110 m × 22 m. Environmental parameters (temperature and humidity) are locally monitored with a weather station, providing additional data.

It is generally impractical to install a wired network, so gateways utilize 4G (LTE-Long-Term Evolution) cellular connections to transmit the collected data to the central system’s database. All received data are stored for long-term evaluation and short-term reports (e.g., overall status, warnings, predictions) are provided through web and mobile interfaces.

### 2.3. Mechanical Design and Power Source

Bolus technology is primarily used for veterinary purposes, particularly for the long-term release of trace elements, resulting in standardized sizes and applicators. This standardization ensures that inserting a bolus is a straightforward procedure for veterinary professionals. The target size of the bolus sensor must accommodate its internal components, with the battery being the most significant factor. To maximize the power budget, a non-rechargeable Lithium D-type battery can be used, dictating the form factor of the bolus.

Using rechargeable batteries was also considered, but rejected. First, recharging is problematic. Wireless energy transfer exists, but impractical for this application. Energy harvesting devices would either produce an inefficient amount of power (e.g., thermoelectric or Radio Frequency radiation based), or make the final design very error-prone (e.g., mechanical ones).

Second, the lifetime of rechargeable batteries is much shorter than non-rechargeable ones, especially when they are charged and discharged with very low currents and kept almost fully charged. Non-rechargeable batteries, on the other hand, can serve for extended time, when discharged with small currents.

The lower size limit of the bolus is the experimental size at which the device no longer remains in the rumen, but passes through the digestive system or returns to the esophagus during rumination, so it is not worth reducing the size of the device too much. In practice, it is worth choosing the size of the device in such a way that it is compatible with one of the already tested and marketed applicators and can be easily administered with it. Based on these constraints and the available applicator sizes the diameter of the bolus is chosen to be 40 mm. The applicator does not give a hard limitation on the length; 120 mm was chosen.

Choosing the material for the enclosure of the bolus sensor requires careful consideration of durability and waterproofness. POM-C (polyoxymethylene) industrial plastic is an excellent choice, offering robust protection against the harsh conditions within the rumen as it has a certificate for usage in the food industry [47]. The photo of the applicator and the enclosure of the bolus sensor is shown in Figure 2.

### 2.4. Sensors

It is essential that the applied sensors provide measurements that can be utilized to derive the requisite characteristics and alerts. Furthermore, the sensors must be accessible, low-power, durable, compact, and cost-effective. The device does not contain expensive components, so the cost of the prepared sensor is comparable to sensors already on the market, for example, a neck transponder. Most of the relevant characteristics pertain to some form of movement or physiological change. To capture these, a MEMS (Micro Electronic Mechanical Systems) accelerometer and/or gyroscope can be employed. These sensors are effective for monitoring various types of movement, providing valuable insights into animal behavior and health. Additionally, a temperature sensor is crucial, as it can deduce several vital functions and health indicators from temperature data.

The use of accelerometers (and/or gyroscopes) and temperature sensors is well-documented in both commercial [16,17,18] and experimental bolus sensors [22,34,35]. Another sensor frequently mentioned in the literature is the pH meter, which helps track the digestion process inside the rumen. However, pH meters typically do not survive for several years, limiting their long-term utility. Therefore, this sensor modality was omitted from the design.

### 2.5. Communication Design

The largest energy consumer in the designed bolus sensor is clearly the communication module, making it imperative to design the energy plan around this factor. The primary constraint on one side is the battery capacity, while on the other side, the objective is to transfer as much data as possible to obtain fine-grained data.

LoRaWAN utilizes single-hop communication, which, when combined with a periodic sensing strategy, offers an easily manageable method. Measurement data are sent via uplink (sensor to gateway) packets. To ensure timely alerts, the system must maintain an expected latency of no more than 2–3 h. To tolerate data loss and enable the system to generate alerts from multiple measurements, sending data hourly represents a good compromise.

LoRa allows a maximum packet size of 51 bytes within the optimal spreading factor range (SF10, SF11, and SF12), with bitrates of 980, 440, and 250 bps, respectively. The corresponding transmission times $t_{tx}$ for these packets are 416 ms, 927 ms, and 1632 ms. A typical LoRa chip consumes between 80 and 120 mA during transmission. For the calculation, the worst case active current $i_{tx}$ = 120 mA and the longest transmission time will be used. Such a system always has a non-zero standby current. With typical settings, the standby current $i_{st}$ of the microcontroller, the sensors and the radio is estimated to be lower than 100 μA. The capacity of a D-type non-rechargeable lithium thionyl chloride (Li-SOCl2) battery is *C* = 19 Ah. The following calculations will use the SF12 spreading factor, which gives the best link budget and results in a worst-case approximation in terms of battery lifetime. The duty cycle of the radio can be calculated as follows.
(1)$$\begin{array}{cc} d & =\frac{t_{tx}}{t_{period}}, \end{array}$$
where $t_{period}$ denotes the sending period time (1 h) and $t_{tx}$ is 1632 ms for SF12. These settings result in $d=0.45\cdot 10^{-3}$. The lifetime can be calculated as follows: (2)$$\begin{array}{cc} t & =\frac{C}{d\cdot i_{tx}+\left(1-d\right)\cdot i_{st}}. \end{array}$$

Using the slowest spreading factor (SF12) with a non-rechargeable Lithium D-type battery, the estimated lifetime for the device, considering only radio communication is approximately 14 years. This calculation provides a solid foundation for the energy budget, ensuring long-term operation.

Additionally, LoRa supports downlink (gateway to sensor) messages, which can be effectively used to adjust the parameters of the preprocessing algorithm or the system. This capability allows for dynamic reconfiguration and optimization, enhancing the flexibility and responsiveness of the bolus sensor system.

## 3. Results

### 3.1. Hardware

#### 3.1.1. Experimental Bolus

The development of the preprocessing algorithms to be executed on the microcontroller of the final bolus necessitated the utilization of a device (the experimental bolus) that permitted the aggregation of raw measurement data.

This bolus version is based on an nRF52840 Dongle, which contains a complex System on Chip (SoC) solution by Nordic Semiconductor (Trondheim, Norway). The microcontroller part of the chip contains a 64 MHz Cortex-M4 core, 1 MB Flash and 256 KB RAM. This board has a convenient USB connector, that enables convenient firmware updates debugging during development. The SoC contains a Bluetooth 5.3 compatible module, which provides another communication possibility for further firmware modifications and testing after sealing the device.

Although the SoC has versatile wireless capabilities, another radio chip was necessary to use for data communication. All protocols offered by the chip use the 2.4 GHz band, which is absorbed by the water. The device uses a Würth (Künzelsau, Germany) AMB3626-M module, which implements a 169 MHz wM-Bus protocol with 4.8 kbps. This module was employed in continuous mode to handle the high data rate necessary for capturing detailed measurements. Note that LoRa is not suitable in this situation, due to its regulated duty cycle.

The acceleration and the temperature are measured by a LIS2DTW12 MEMS digital output dual motion and temperature sensor. The physical structure of the three modules and the battery is shown in Figure 3a.

The energy source of the experimental bolus sensor was the same D-type lithium battery as the final version, which achieved a lifetime of 15 days. The system continuously checks the battery’s charge and sends this information hourly to the server. Sensor malfunctions are not systematically checked, but the missing measurements are detectable on the server side.

The main component of the prototype gateway was the Raspberry Pi single-board computer (SBC). During the experiment no real-time data forwarding was required; the gateway only recorded the received data frames to a high-capacity SD card, which was collected at the end of the experiment. Occasional data access was still possible through an Ethernet or WiFi link. The data frames from the bolus sensor were received by the same AMB3626-M module. The photo of the gateway is shown in Figure 3b.

Concurrent with the data collection experiment, the instrumented animal was also subjected to visual inspection with continuous camera recording, and its heart rate (HR) was monitored. An additional tilt sensor was attached to one of the animal’s legs to accurately identify standing and lying periods. All measurements were synchronized. This approach facilitated the identification of events and provided an alternative means of validation assessment during the development phase.

This comprehensive data collection enabled the identification of patterns in the raw data, which was essential for the development of the preprocessing algorithms. These patterns provided critical insights into the animal’s behavior and physiological state, forming the foundation for robust and efficient on-device data processing in the final bolus system.

#### 3.1.2. Final Bolus Design

The final bolus uses an nRF52840 microcontroller as its base unit (see Figure 4. This device has 1 MB flash and 256 KB RAM, which provides plenty of space for processing algorithms and general storage. It is based on a 64 MHz Cortex-M4 CPU core. This chip has a multifunctional 2.4 GHz radio communication core, too, which is Bluetooth Low Energy 5.3 capable. This enables firmware updates after sealing the bolus, but it cannot be used when the bolus is inside the rumen.

The LoRa communication uses an AcSiP S62F chip. The spreading factor can be set between SF10 and SF12 with the default setting of SF12, which offers the largest possible link budget. The bolus sends a data packet of 51 bytes (the maximum allowed packet size for the used spreading factors) in each hour. The data packet contains the following data:Packet IDCompensated body temperatureDrinking counterCumulated activity countersPeak activity valuesRumination counterHeart rate valuesMaintenance data (firmware version, system status, battery voltage)

### 3.2. Software

#### 3.2.1. Embedded Firmware

The architecture of the embedded firmware and the main components of the hardware are shown in Figure 5. The internal peripherals of the microcontroller (MCU) can be accessed directly by the firmware. The Timer can initiate periodic system events. The analog–digital converter (ADC) is used to measure the battery voltage. The General Purpose IO (GPIO) and Serial Peripheral Interface (SPI) are used to communicate with the external peripherals (the accelerometer, temperature sensor and LoRa radio).

The firmware consists of two layers. The lower Services layer communicates with the internal and external peripherals and provides abstract interfaces to the Application layer. The Measurement Scheduler initiates the regular measurement events. The accelerometer is used at 12.5 Hz, while the temperature sensor is used at a 1 Hz sample rate. The Buffer Organizer organizes the message buffer and provides preallocated space for each computed data. The Radio handler initiates the message sending once each hour. The Settings manager handles the downlink (gateway to sensor) messages, which can contain parameters for the processing algorithms. This module also stores the settings in flash memory and reloads them after reset events. Finally, this layer contains a Self Diagnostics module, which monitors the integrity of the system and generates a reset in case of a soft lockup.

The Application layer runs the different detection algorithms (Drinking detection, Motion activity detection, Heart rate detection, Rumination detection), and the firmware support for the Thermal shock/fever and the Oestrus prediction algorithms.

The preprocessing algorithms can be considered as a special kind of (lossy) data compression. With a fixed measurement setup, the compression ratio can be calculated as follows.
$$r_{c}=\frac{d_{o}}{d_{i}},$$
where $r_{c}$ is the computed compression ratio, $d_{o}$ is the number of output bytes and $d_{i}$ is the number of input bytes, during an arbitrary amount of time. For the calculations, a one hour period will be used. In this case $d_{o}=51$ bytes is the size of a single radio packet and $d_{i}$ is the amount of data measured by the sensors during this period. With 12.5 Hz sample rate, three axes and 2 bytes per axis the accelerometer generates 3600·12.5·3·2 = 270,000 bytes. The temperature sensor does a measurement every 10 min (6 times each hour) and it generates 2 bytes each time, which means 12 bytes per hour. As a result, $d_{i}=$ 270,012 bytes and $r_{c}\approx 1.9\cdot 10^{-4}$.

#### 3.2.2. Drinking Detection Algorithm

Figure 6 shows a 24 h temperature record (red chart). Real drinking events are marked with asterisks, based on the video footage. In all cases, an abrupt temperature drop can be observed in the rumen temperature due to the high volume of added cold water. The temperature then slowly returns to the original level. Previous research on the detection of drinking has been published in [33].

The bolus measures the rumen temperature *t* every 5 min. The algorithm compares consecutive temperature samples and counts the events, where the temperature decreases and the difference between the actual temperature *t* and the previous one $t_{last}$ is at least th. To prevent false double detections a *w* window is defined; at most one detection is allowed in the window. The default value for *w* is 10 min (two samples). The default value of th is 2 °C, but it can be fine-tuned based on the temperature of the drinking water. Based on the noise characteristics of the temperature sensor th cannot be lower than 1 °C. The drinking detection algorithm is shown in Algorithm 1.  
**Algorithm 1:** Drinking Detection **Initialization**          r=0       *remaining samples to the next possible drinking event*         $t_{last}$          *last measured temperature*         c=0       *drinking event counter* **Input**          *t*              *temperature* **For each sample:**  1     **if** $t<t_{last}-th$ **and** r=0 2            c=c+1 3            r=w 4     **if** r>0 5            r=r−1 6     $t_{last}=t$

#### 3.2.3. Motion Activity Detection Algorithm

The signal measured by the accelerometer primarily consists of two main components. The first component is the gravitational force, which is distributed across the three axes (x, y, and z) depending on the orientation of the bolus and the sensor. The vector sum of this component is approximately 1 G. Over short time intervals, this component can be considered constant; however, it varies as the orientation of the bolus changes. For a 60 min raw record, see Figure 7.

The second component is the acceleration of the bolus caused by various movements. These movements can be categorized into three types:General Motion: The motion of the animal produces larger, mostly aperiodic values. These variations occur due to activities such as walking, running, or sudden movements.Heartbeat: The effect of the heartbeats produces smaller, periodic changes in the accelerometer readings. These subtle variations are indicative of the animal’s cardiovascular activity.Rumination: Rumination creates a very specific pattern, which is associated with the repetitive chewing and digestive processes of the animal.

By analyzing these two main components—gravitational force and movement-induced acceleration—the accelerometer provides valuable insights into the animal’s orientation, activity levels, and physiological state.

Two distinct algorithms have been implemented to detect motion activity. The first algorithm is a peak detector, which identifies the highest peak in the absolute value of the acceleration values within a specified time interval. The search is performed in non-overlapping 5 min blocks. The maximum value is categorized into one of 16 predefined categories using thresholding. Each category is encoded into a 4-bit value for efficient data transmission and processing.

The second algorithm for activity monitoring first filters out the effect of the gravity. It uses an exponential averager for each sample: (3)$$\begin{array}{cc} \hat{x} & =\alpha \cdot x+\left(1-\alpha \right)\cdot \hat{x}\\ \hat{y} & =\alpha \cdot y+\left(1-\alpha \right)\cdot \hat{y}\\ \hat{z} & =\alpha \cdot z+\left(1-\alpha \right)\cdot \hat{z}, \end{array}$$
where *x*, *y* and *z* are the measured acceleration values for each axis, α is the parameter of the exponential averager, $\hat{x}$, $\hat{y}$ and $\hat{z}$ are the filtered acceleration values.

Then the i intensity value is calculated, which is the distance between the filtered and the unfiltered acceleration vector.
(4)$$i=\sqrt{{\left(x-\hat{x}\right)}^{2}+{\left(y-\hat{y}\right)}^{2}+{\left(z-\hat{z}\right)}^{2}}$$

Then, the i intensity value is smoothed with another exponential averager: (5)$$\hat{i}=\beta \cdot i+\left(1-\beta \right)\cdot \hat{i},$$
where β is the parameter of the exponential averager and $\hat{i}$ is the smoothed intensity value.

Finally, the intensity is categorized into three categories with two thresholds and the counter for that class is incremented. In order to reduce the amount of data transmitted, it is possible to further quantize the value of the counter prior to its transmission.

#### 3.2.4. Rumination Detection Algorithm

The rumination detection algorithm is based on the i intensity value (calculated by the motion activity detection algorithm) with additional smoothing and basic envelope detection, resulting in the ${\hat{i}}_{e}$ smoothed envelope intensity value, which is calculated as follows: (6)$$\begin{array}{cc} \mathrm{if}\;i>{\hat{i}}_{e}:{\hat{i}}_{e} & =i\\ \mathrm{else}\;{\hat{i}}_{e} & =\beta \cdot i+\left(1-\beta \right)\cdot {\hat{i}}_{e}, \end{array}$$

Figure 8 shows an hour of smoothed envelope intensity data, where a 16 min long rumination period (between minutes 22 and 38), preceded and followed by other activities, is shown. During rumination longer inactive periods and shorter active ones are happening after each other, regularly. The algorithm finds this periodic pattern and cumulates the length of the proper time periods. The exact criteria are specified as follows:The active state starts, when ${\hat{i}}_{e}$ passes the $th_{H}$ high threshold (default value: 3500).The passive state starts, when ${\hat{i}}_{e}$ falls under the low threshold (default value: 6000).The active period is accepted when its length falls into a predefined $\left[t_{act,min},t_{act,max}\right]$ interval.The active-passive period pair is accepted when their total length falls into a predefined $\left[t_{int,min},t_{int,max}\right]$ interval and the relative difference between the *k* consecutive interval lengths is lower than $\Delta_{tim}$. The default value for $t_{int,min}$ and $t_{int,max}$ are 50 and 100 s, or 625 and 1250 samples, respectively. The default value for *k* and $\Delta_{tim}$ are 5 and 0.2, respectively.

This pattern is recognized with Algorithm 2.    
**Algorithm 2:** Rumination Detection **Initialization**          s=0       *activity state*         $t_{0}=0$      *sample number of the start of the last active period*         $t_{1}=0$      *sample number of the end of the last active period*         $t_{2}=0$      *sample number of the end of the last passive period*         L=[]      *buffer for period lengths, stores k values, FIFO type *         c=0       *cumulated rumination*         o=0       *ongoing rumination* **Input**          *t*              *sample number*         ${\hat{i}}_{e}$            *enveloped intensity value* **For each sample:**  1      **if** $s=0\wedge {\hat{i}}_{e}\geq th_{h}$ 2             s←1 3             $t_{2}\leftarrow t_{0}$ 4             $t_{0}\leftarrow t$ 5             $l\leftarrow t_{2}-t_{0}$ 6             **if** $l\notin [t_{int,min},t_{int,max}]$ 7                    o=0 8                    L=[] 9             **else** 10                  append L,l 11                  **if** $length\left(L\right)=k\wedge \frac{min(L)}{max(L)}>1-\Delta_{tim}$ 12                         **if** o = 0 13                                c = c + sum(L) 14                                o = 1 15                         **else** 16                                c = c + l 17    **if** $s=1\wedge {\hat{i}}_{e}\leq th_{l}$ 18           s←0 19           $t_{1}\leftarrow t$ 20           $l\leftarrow t_{1}-t_{0}$ 21           **if** $l\notin [t_{act,min},t_{act,max}]$ 22                  o=0 23                  L=[]

The algorithm stores the activity state in variable *s* (0 means inactive, where 1 means active state). The $t_{0}$, $t_{1}$ and $t_{2}$ variables store the time, then an active period started, ended (and at the same time, a passive one started), that one ended. This means, that, for a full active-passive period pair, the length of that period is $t_{2}-t_{0}$, while the length of the active part is $t_{1}-t_{0}$. *L* is a FIFO-type buffer, which stores the length of the full periods *c* stores the cumulated rumination value, and finally *o* tracks the ongoing rumination state.

The algorithm runs for each incoming ${\hat{i}}_{e}$ value and the other input *t* is the actual sample number. From this value the *s* state is tracked by lines 1–2 and 17–18. Lines 3–5 and 18 update the $t_{0,1,2}$ variables.

For full periods, line 5 calculates the length and line 6 checks the related condition. The negative case means the interruption of the pattern; *o* is nulled and the content of *L* is dropped (lines 7 and 8). When the condition is true, then the actual period length gets appended to *L* (line 10) and then the algorithm checks, if *k* full periods are collected in *L*, and if the similarity of these numbers is within the required tolerance (line 10), rumination is confirmed. When it is a newly detected one (o=0, line 12), the sum of the elements of *L* is added to the counter *c* and *o* is set to 1 (lines 13 and 14). When there was rumination already, *l* is simply added to *c* (line 15). Lines 21–23 drop the periods with too short or too long active parts.

#### 3.2.5. Heart Rate (HR) Measurement Algorithm

To the best of our knowledge, no other HR-detecting bolus has been developed yet. However, several papers have addressed the importance of HR measurements in dairy cattle farming [42,43]. To date, there have been no studies on the heart rate detection of cows. However, a substantial body of literature exists on this topic in relation to humans [48]. The findings of these studies indicate that it is challenging to perform this task accurately and effectively. In the context of human use, devices typically operate in conjunction with electrocardiogram (ECG) signals or determine heart analyzing data obtained from pulse oximetry or acceleration measurements [48]. There are numerous publications on data processing methodologies, which often use computationally intensive signal processing methods due to the significant noise present in the measurement [48].

Here, we aim to estimate HR from 3D accelerometer activity measurements of the cattle rumen bolus. In the bolus 3D acceleration data are measured using an accelerometer with a sampling frequency of 25 Hz. Figure 9 shows a real acceleration data segment and also the real heartbeat events, measured by an ECG device. Although the heartbeats are slightly visible in some cases, the animal’s locomotor activity is strongly superimposed on the curve, and the IBI is hard to detect.

In our previous research, we have explored effective and sophisticated methods for cow HR detection, but these algorithms are computationally demanding, presenting a challenge for implementation in the bolus sensor [49,50]. In the development of the current algorithm, it is of significant importance to consider that the primary computations must be conducted on the microcontroller. Following this, a small amount of data can be transferred to the server, where additional computations can be performed. The block diagram of the algorithm is shown in Figure 10. The computations performed on the bolus sensor are illustrated on the left-hand side of Figure 10, while the server-side computations are shown on the right-hand side.

The accelerometer produces 3D measurements $ACC_{x}$, $ACC_{y}$, and $ACC_{z}$, as shown in Figure 10. The samples are collected in three buffers of length 75, encompassing a total of three seconds’ worth of data. For the purpose of detection, the two axes exhibiting the greatest acceleration are selected, as the cow’s heart is situated in close proximity to the vertical axis, making it more susceptible to acceleration due to cardiac activity. In the following steps, the computations are performed for the two selected axes, referred to as *A* and *B*.

In the next step, signal conditioning is performed, including baseline removal and low-pass filtering. In the peak detection phase, local maxima and minima are sought in both selected axes. The outputs of the peak detection blocks are the interval lengths between the maximum points ($I_{A,max}$, $I_{B,max}$) and the interval lengths between the minimum points ($I_{A,min}$, $I_{B,min}$) on both axes. It has been described in the literature that motion artifacts can generate spurious periodicities [48]. In order to reduce the number of false detections, the median value of the detected period lengths is calculated. Thus, the output of the bolus-side computation is the four estimated period lengths, which should ideally all be equal to the actual IBI.

On the server side, post-processing is conducted by a Multi-Layer Perceptron (MLP) artificial neural network comprising four input and eight output neurons. Consequently, it can classify IBI values into eight categories between 50 and 130 bpm [50].

The neural network was trained using parallel ECG data as labels. The control measurements were conducted using a Polar ECG device. The distribution of mean absolute error values for the detection algorithm is illustrated in Figure 11. The developed algorithm exhibits a 10% error rate, which is inadequate for medical or veterinary measurement but may offer useful insight in agricultural contexts.

#### 3.2.6. Fever and Thermal Shock Detection Algorithm

Fever and thermal shock detection is still under ongoing research. While the primary data collected by the bolus sensor provide a valuable foundation for identifying potential fever and thermal stress, they are not sufficient on their own. Additional contextual data, such as environmental factors, along with long-term data processing, are essential for accurate detection. Therefore, a reliable alert system for fever and thermal shock can only be effectively implemented on the server side.

The bolus does not directly measure the body temperature of the animal. Drinking affects the measurements in two ways. First, the rumen temperature is directly reduced by drinking. Second, drinking reduces the heat released during fermentation by decreasing the enzyme activity of fermenting bacteria. Rumen fermentation is also related to rumen temperature, as increased temperature refers to increased fermentation in the rumen, which releases heat through exothermic chemical reactions and causes a temperature rise of 1–2 °C. These compensatory calculations are, in turn, proposed to be performed on the server side by a machine learning algorithm. Barn temperature, drinking information and rumination data could provide useful supplementary information for the compensation. Another important condition of the compensation is a significant amount of data is collected by observing large numbers of boluses over a longer period of time. For direct temperature monitoring, the test is performed most commonly by measuring the temperature of the animal in the rectum. There are extensive studies on the relationship between rumen temperature and core body temperature, which can be used as a good starting point [31,35,43].

The compensated temperature serves as the fundamental data for a sophisticated fever alarm, which can be achieved through simple thresholding. These temperature data are also crucial for detecting heat shock conditions [29]. It is important to consider environmental factors such as ambient temperature and humidity when setting these alarms, as they can significantly impact the animals’ thermal regulation.

#### 3.2.7. Oestrus Prediction Algorithm

Another ongoing area of research is the detection of oestrus. Oestrus detection can be solved on the server side by using a machine-learning solution. Oestrus is associated with characteristic changes in activity and body temperature. By recording body temperature and movement activity every hour, using a 24-h time window of these data, a deep neural network can be used to detect oestrus. In order to achieve a universal algorithm, it is necessary to take into account the local characteristics of the farm, the differences in farming, and the individual characteristics of the animals. To do this, it is practical to record a daily activity profile for each animal, taking into account about 10 days of data, and the input to the detection algorithm will be the deviation from the activity profile. Seasonal trends must also be considered in the case of temperature, so the deviation from the average of the last 3 days at the same hour should be used. Convolutional neural networks and different recurrent neural network solutions were utilized in our experiments.

## 4. Discussion

This section presents a comprehensive overview of the principal challenges associated with the development of a rumen bolus. The novel features of the device include an extended operational lifespan, the capacity for lifelong monitoring, on-device data processing to yield physiologically meaningful insights, and, in particular, the ability to obtain heart rate data from the sensor. Based on the literature comparison, it is striving to be at the forefront of sensors in the market if it is marketed as a product. It is capable of measuring all the characteristics defined in [44], and it can also detect rumination and HR levels. The connection between HR and stress has been shown in several studies [42,51,52] making HR detection an important feature. Thus, stress can be detected through the HR, which affects the animal’s reproductive capacity and milk production [52]. The HR can also be helpful in monitoring the state of health and calving.

The principal design constraints were the form factor and the obligatory requirement for longevity. In the paper, we outlined the design and implementation of the actual hardware, as well as the design and implementation of the firmware, with special emphasis on the communication and the measurement strategy. It is our conclusion that the optimal strategy is to conduct the primary evaluation of the data within the firmware, with the bolus transmitting already physiologically interpretable, compact data via the gateway to the server. In our evaluation, the primary characteristics that should be analyzed inside the bolus are the number of drinks, the rumen temperature, the activity of movement, the rumination time, and the heart rate. The calculation of these characteristics can be performed using a variety of algorithms, some of the on-device solutions are presented in this paper. It appears that it is possible to evaluate data concerning the quantity of nourishment ingested and the duration spent in either a standing or recumbent position. These parameters may serve as potential avenues for further research.

We propose that the server should subsequently undertake a post-processing of the primary characteristics that were transmitted. The following fields may benefit from the implementation of post-processing techniques:Through post-processing of the heart rate data, it is possible to obtain valid data free of the influence of false periods. Furthermore, data processing should incorporate data from the sensor in the stable, as well as data from the animal administration systems, including expected estrus, medication, and other relevant variables.An elevated temperature in the rumen may also indicate increased fermentation activity. However, when this is considered alongside the time spent ruminating and the temperature in the barn, valuable information about fever can be obtained. The same data can be used to infer heat stress by comparing them with stable humidity and temperature.The detection of estrus on the server side is feasible based on the analysis of movement activity and temperature data.The combination of activity and drinking data enables the detection of calving events. Complex monitoring of the animal may indicate the presence of additional health issues, including disease and lameness, through the observation of altered behavioral patterns.

The sensor’s steady positioning renders it a dependable data collection instrument for livestock farming, facilitating continuous monitoring in the stable. In the pasture, however, it is often not feasible to establish an adequate radio communication infrastructure due to the limited area that can be covered by gateways, which are subject to attenuation from the animal’s body. Consequently, external sensors are now considered a superior option in pasture. Nevertheless, for grazing animals, rumen bolus sensors can be developed that send information intermittently, for example at milking for dairy cows or at watering for beef cattle. This information can then be transmitted by gateways placed at the given location. The prospective future development options for the rumen bolus are as follows:At present, there is no pH measurement sensor that allows for long-term monitoring. The introduction of such a device would confer a significant advantage to rumen boluses over other sensors, given the paramount importance of rumen pH measurement and the inability to achieve it through alternative means.The necessity for a more comprehensive understanding of the quantity of greenhouse gases emitted by cows and the means of controlling this level in animal husbandry is increasing. Consequently, it is anticipated that in the future, the development of boluses for the detection of methane gas will receive significant attention.The issue of feed utilization is of significant importance, and it is reasonable to anticipate the advent of related measurements in the future.Digital twinning is a simulation and visualization environment that provides real-time data from the actual operation of systems, allowing for the visualization and monitoring of these systems from a multitude of perspectives. Such systems are currently utilized in industrial contexts, yet their application in agricultural settings is still in its infancy. However, their advent is anticipated [53].

The management of data generated in animal husbandry represents a significant challenge and one of the most crucial issues in precision livestock farming. In the context of farm operations, data pertaining to the health and husbandry of individual animals are already extensively collected, managed, and utilized by dedicated software applications. Additionally, sensors, such as external sensors and rumen boluses, gather and store a considerable amount of data, which are generated independently of the aforementioned data. The data are typically collected in isolation, and data transfer between systems can occasionally occur through individual interfaces. The value of the systems would be significantly enhanced if these applications could communicate with each other continuously and allow for integrated data collection. It is probable that data communication standards like those used in industry would be needed, and it would be worthwhile to explore whether existing industry standards could be adapted for PLF. It is anticipated that this will culminate in a future in which all dairy cattle are monitored and integrated into a highly automated, environmentally friendly, and animal welfare-friendly sustainable PLF system.

## Figures and Tables

**Figure 1 sensors-24-06976-f001:**
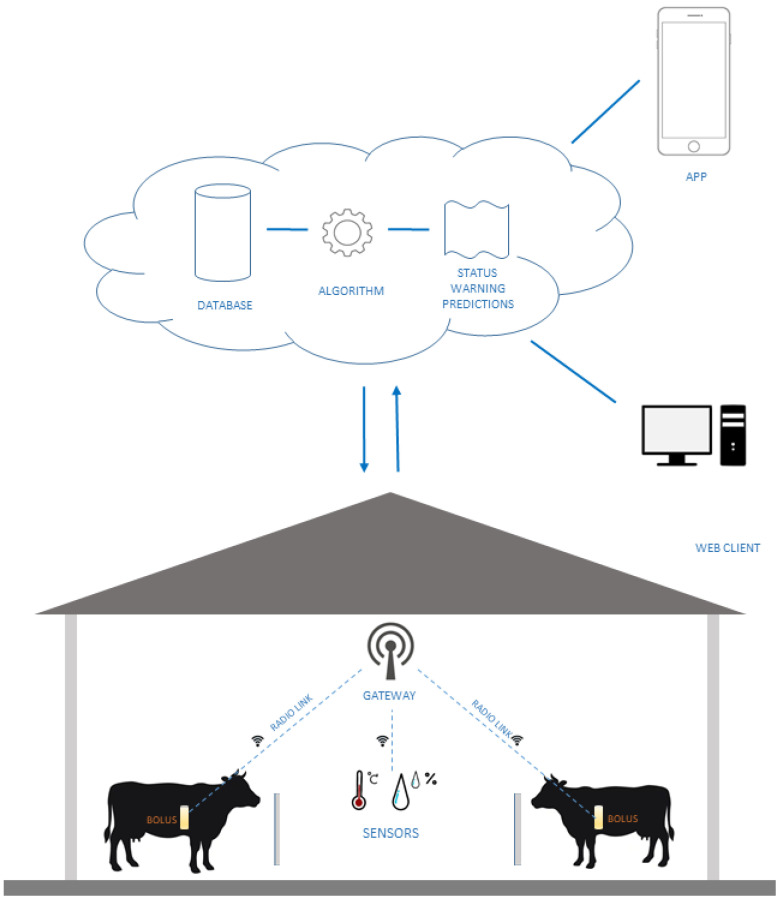
The architecture of the bolus sensors system.

**Figure 2 sensors-24-06976-f002:**
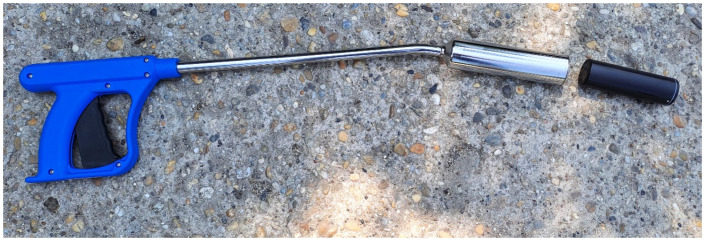
The applicator with the waterproof enclusure of the bolus.

**Figure 3 sensors-24-06976-f003:**
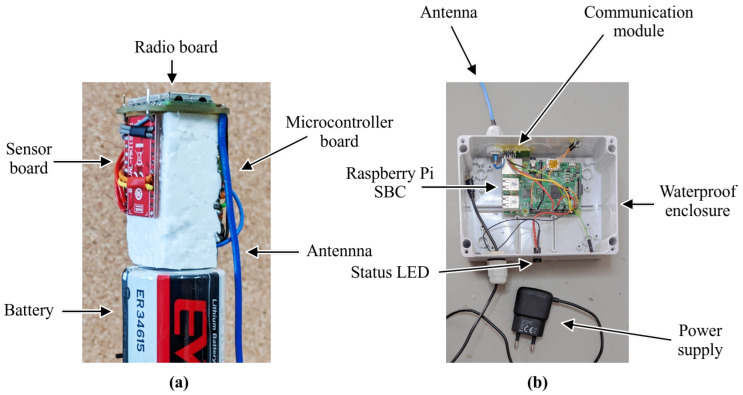
The experimental bolus sensor (**a**) and the gateway (**b**).

**Figure 4 sensors-24-06976-f004:**
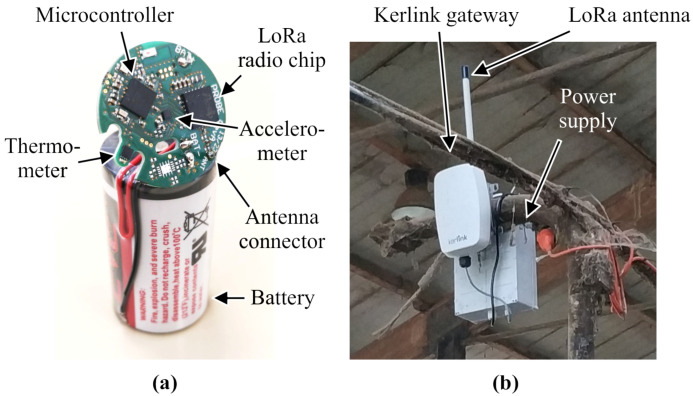
The printed circuit board of the bolus with the high capacity battery (**a**) and the gateway installed under the roof of the barn (**b**).

**Figure 5 sensors-24-06976-f005:**
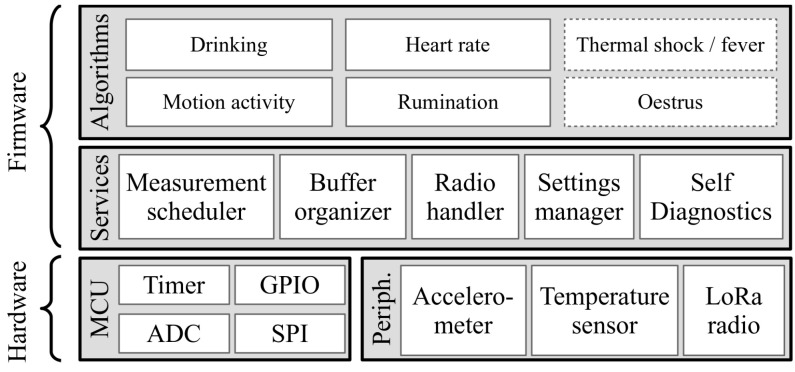
The architecture of the firmware with the hardware components.

**Figure 6 sensors-24-06976-f006:**
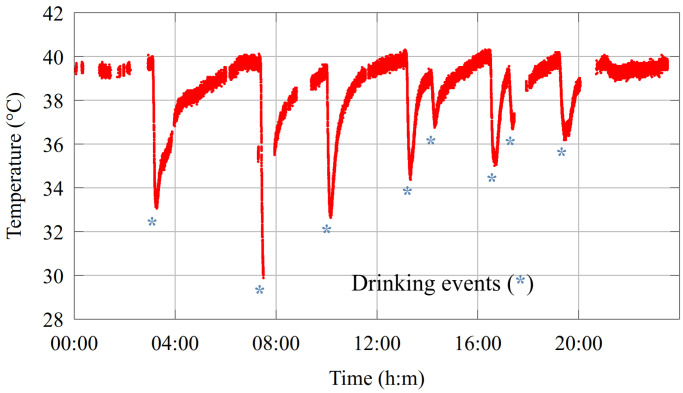
Native rumen temperature measured every 5 min. The drinking events observable in the video footage are marked with asterisks. A total of eight drinking events are visible on the chart.

**Figure 7 sensors-24-06976-f007:**
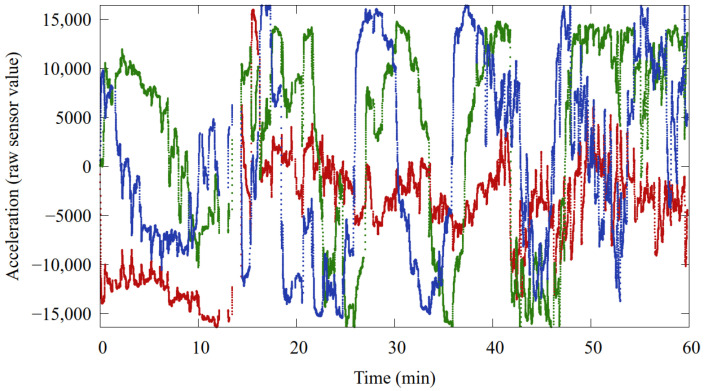
The typical activity pattern. Red, green and blue traces denote the raw X, Y and Z acceleration values.

**Figure 8 sensors-24-06976-f008:**
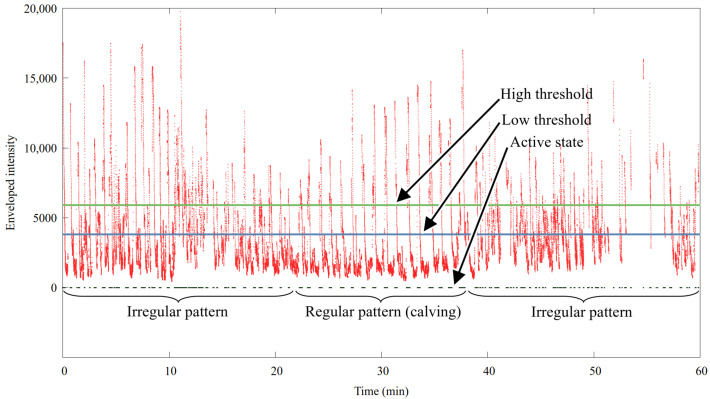
The smoothed envelope of the intensity pattern with a 16 min long rumination period. The high and low thresholds used by the detection algorithms are marked with a green and a blue line, respectively. The active state is marked with a black trace. Note that the absence of the black marking means passive state.

**Figure 9 sensors-24-06976-f009:**
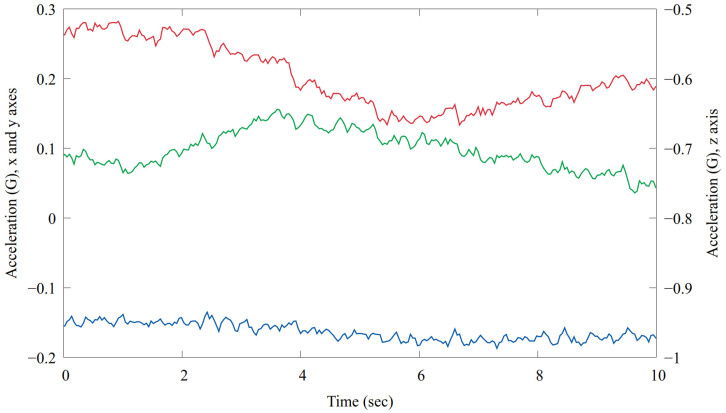
A 10 s raw record of the accelerometer. The red, green and blue plots denote the *x*, *y* and *z* axes. Note that the *x* and *y* axes are plotted on a [−0.2;0.3] range, while the *z* axis is plotted on a [−1;−0.5] range. The heartbeats measured by the ECG are marked with gray dashed lines.

**Figure 10 sensors-24-06976-f010:**
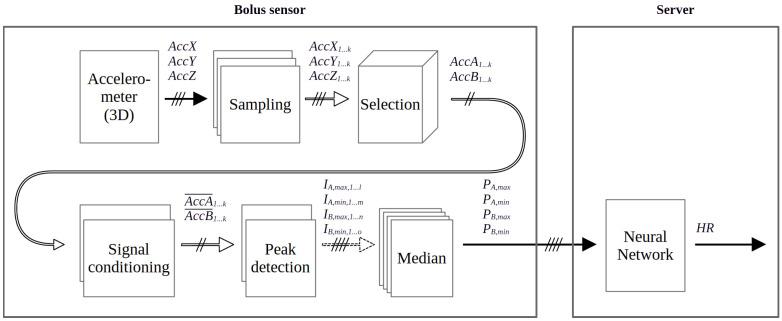
The block scheme of the HR estimation algorithm with the processing steps implemented inside the bolus (**left side**) and on the server (**right side**).

**Figure 11 sensors-24-06976-f011:**
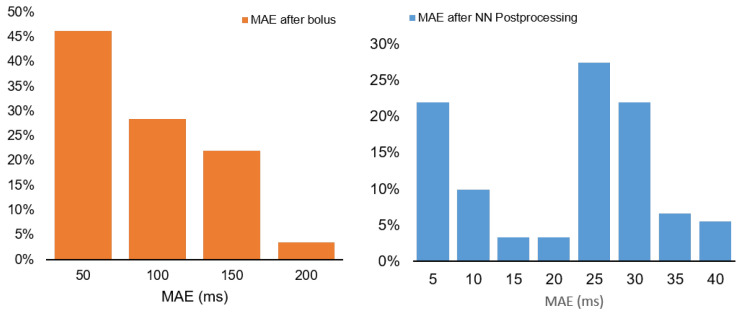
The mean absolute error(MAE) of heart rate detection in the bolus and after postprocessing on the server. The mean heart rate of the dairy cattle is in range 700–800 ms, so the error is in 10%.

## Data Availability

The datasets collected and analyzed during the current study are available in the following repository on GitHub: https://github.com/Bolus-2021/Bolus_2024_data (accessed on 13 October 2024). This repository contains all relevant data and supplementary materials required to reproduce the findings of this study.

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
