# Peer review of "In-Depth Development of a Versatile Rumen Bolus Sensor for Dairy Cattle"

_sensors, 2024, doi:10.3390/s24216976_

Round 1
Reviewer 1 Report
Comments and Suggestions for Authors
The article under review discusses the development of a rumen bolus sensor for dairy cattle, aimed at improving health monitoring and oestrus detection in precision livestock farming (PLF). The paper addresses a timely issue in PLF. The authors provide a detailed breakdown of the design process, covering mechanical design, sensor modalities, power supply, and communication protocols. While this is commendable, the work could benefit from deeper exploration of some components, particularly in comparing alternative technologies. In summary, the paper is a strong contribution to the field, further research and comparative analysis with other solutions would enhance its impact and applicability in real-world settings.
Important: There is no mention of approval from an Animal Ethics Committee.
Detailed review:
ABSTRACT
- The abstract could include a brief statement on the validation methods used (e.g., field testing).
- It could also mention whether the paper introduces any novel solutions or comparative analysis of existing technologies.
- L2-4: The sentence is somewhat vague. While it emphasizes the importance of the technology, it lacks specificity regarding why these sensors are valuable in precision agriculture. A clearer definition of their purpose (e.g., monitoring health, detecting oestrus, etc.) could improve the reader’s understanding.
KEYWORDS
- L9-10: Avoid using terms that are already in the title, such as "dairy cattle" and "rumen bolus." You should replace them with more specific terms, broader terms, or synonyms. For "dairy cattle," "precision livestock farming" is a good alternative. For "rumen bolus," alternatives could include "intraruminal device," "gastrointestinal sensor," or "in vivo sensor.
INTRODUCTION
- L12-14: It would be more appropriate to introduce a discussion on Agriculture 5.0 instead of 4.0.
- L38-53: While the historical context and justification for bolus sensors are well-established, there is a lack of clarity regarding the specific research gap this paper aims to address.
- L65-69: Several technologies are mentioned, but a critical discussion is missing regarding how this work differentiates itself or improves upon prior research.
- L65-69: It would be helpful to better outline why their work is essential in the current landscape of bolus sensor technology.
- L54-62: It is important to discuss how their developments in bolus sensor technology contribute directly to solving real-world challenges faced by farmers.
METHODOLOGY
- The authors focused on system design, but there is a lack of clarity regarding the methodological decisions.
- How was the problem scope defined?
- How were the requirements specified? The specification seems entirely empirical. It would be better to follow a well-known approach like IEEE 29148 or ISO/IEC/IEEE 15288.
- Who were the stakeholders involved in defining the requirements?
- How were the scenarios defined?
- How were the experimental units selected?
- How was the field testing planned and specified?
- How was the solution validated?
- Can the proposed solution be compared with similar solutions?
- L141-146: Lacks details.
- L154: What are the dimensions of the barn?
SYSTEM DESIGN
- Except for the communication aspects, section 2.1 lacks references.
- There is no mention of approval from an Animal Ethics Committee.
Figure 1: "Algorhithm"
Some acronyms were not introduced: LTE, POM-C, MEMS, LoRaWAN.
L157. You should define what is "necessitating strategic placement of gateways".
L169. There is little discussion on the energy efficiency trade-offs between non-rechargeable and rechargeable batteries or alternative power sources such as energy harvesting, which has gained attention in remote sensor applications.
L175. The discussion could be enhanced by providing data or references to support the material's performance in similar applications. Additionally, the long-term degradation of the material due to digestive fluids or extreme environmental conditions should be addressed. You should include references to studies that validate the use of POM-C or similar materials in comparable biological environments. Discuss the results of any mechanical stress tests or long-term durability studies to assure readers of the material's suitability.
L194. The text lacks references for field validation data or simulation results to support the claimed communication performance. While the use of LoRaWAN is practical, it would be valuable to compare it with other low-power communication alternatives like NB-IoT or Sigfox, especially in the context of coverage, scalability, and power consumption.
L198. The battery life estimation is well-detailed. However, it would be more robust if supported by real-world usage scenarios or relevant references.
How will the device be monitored for battery degradation or sensor malfunctions?
RESULTS
L227- You should explain more explicitly why each component (e.g., SoC, radio chip, MEMS sensor) was chosen. Include details about trade-offs (e.g., power consumption, cost, signal penetration) for a more rigorous discussion.
L258. The results of validation are not presented. You should include quantitative validation of the system, such as comparing the accuracy of bolus data with the external sensor data. Provide error rates or statistical measures (e.g., correlation, precision, recall) to strengthen the results.
Discuss any anomalies or limitations observed during testing and how these might affect real-world deployment.
L246. You should include a comparison of theoretical battery life with actual performance from field experiments.
L261. You should provide a detailed description of the preprocessing algorithms used (e.g., which techniques or models were applied and why). Include results on how effective these algorithms are in reducing data volume while maintaining accuracy.
L274. While the system records data hourly, there is no discussion on how often this data needs to be transmitted for actionable insights. You should provide a trade-off analysis between data transmission frequency and system responsiveness. Explain how hourly transmissions impact real-time alerting and decision-making.
Address data loss scenarios and how the system can mitigate these (e.g., through data buffering, error correction, or redundancy).
Include metrics for evaluating algorithm performance, such as data compression ratios or computational load on the microcontroller.
Provide a comparison of the proposed system with other commercial or experimental bolus sensors. Highlight the specific advancements this system brings, such as improved data accuracy, longer lifespan, or new data insights.
Further elaborate on the importance of heart rate monitoring and its practical applications in livestock management. Include references to studies that emphasize the role of heart rate in detecting specific health conditions (e.g., stress, illness).
Author Response
We are grateful for your thorough and forward-looking review of our article, and for your valuable comments and questions. We hope that our responses, additions, and revisions to the article will be accepted.
comments 1
ABSTRACT
- The abstract could include a brief statement on the validation methods used (e.g., field testing).
response 1. The mentioned information is inserted.
„We address the issues of conceptual design, overview of applicable sensor modalities, mechanical design, power supply design, applicable hardware solutions, applicable communication solutions and finally the sensor detection algorithms proved in field tests.”
comments 2
- It could also mention whether the paper introduces any novel solutions or comparative analysis of existing technologies.
response 2 The mentioned information is inserted.
„The objective of this paper is to provide a comprehensive overview of the development process of a new sensor development.”
comments 3
- L2-4: The sentence is somewhat vague. While it emphasizes the importance of the technology, it lacks specificity regarding why these sensors are valuable in precision agriculture. A clearer definition of their purpose (e.g., monitoring health, detecting oestrus, etc.) could improve the reader’s understanding.
response 3 The mentioned information is inserted.
„Rumen bolus sensors, which are cutting edge technology and a rapidly developing research field, present an exceptional opportunity for monitoring the health status, physiological parameters, estrus of the animals.”
comments 4
KEYWORDS
- L9-10: Avoid using terms that are already in the title, such as "dairy cattle" and "rumen bolus." You should replace them with more specific terms, broader terms, or synonyms. For "dairy cattle," "precision livestock farming" is a good alternative. For "rumen bolus," alternatives could include "intraruminal device," "gastrointestinal sensor," or "in vivo sensor.
response 4. The mentioned fix is done
comments 5.
INTRODUCTION
- L12-14: It would be more appropriate to introduce a discussion on Agriculture 5.0 instead of 4.0.
response 5. The mentioned fix is done
comments 6.
- L38-53: While the historical context and justification for bolus sensors are well-established, there is a lack of clarity regarding the specific research gap this paper aims to address.
response 6. A new paragraph was inserted with this content.
„Our research aimed to develop a lifelong rumen bolus sensor system, which is capable of measuring many physiological attributes, including temperature, drinking, motion activity, rumination, and heart rate, instead of using multiple sensors. The system is able to process these data with the help of AI on the server side to produce a complex monitoring of animal welfare, health, and farming parameters. The process of creating such a system involves multiple steps, beginning with the creation of hardware, followed by primary measurements, and finally high-level data processing. This paper shows the development of the system, including primary data processing.”
comments 7.
- L65-69: Several technologies are mentioned, but a critical discussion is missing regarding how this work differentiates itself or improves upon prior research.
response 7.
It is not a scientific statement, but from personal hearings was that several farms started to use rumen boluses but it was found to be not enough reliable and finally they stopped using them. Our hope that we can create a really useful sensor system.
comments 8.
- L65-69: It would be helpful to better outline why their work is essential in the current landscape of bolus sensor technology.
response 8.
There is no Heart rate measuring sensor- it is a completely new function.
comments 9.
- L54-62: It is important to discuss how their developments in bolus sensor technology contribute directly to solving real-world challenges faced by farmers.
response 9. A short paragraph was inserted.
„The rumen bolus sensor technology supports sustainable animal husbandry by continuously monitoring animals, reducing the need for human resources for animal husbandry, and supporting the farmer in animal husbandry decisions.”
comments 10.
METHODOLOGY
- The authors focused on system design, but there is a lack of clarity regarding the methodological decisions.
response 10.
Our methodological choices were based on prior expert experience, and we opted for methodologies and technologies that have already been proven to work in research tasks thus far.
comments 11.
- How was the problem scope defined?
response 11.
The problem scope was defined based on the experience of an expert team (veterinarian, farmer, agronomical expert, and engineers).
comments 12.
- How were the requirements specified? The specification seems entirely empirical. It would be better to follow a well-known approach like IEEE 29148 or ISO/IEC/IEEE 15288.
response 12.
The project started by the grant support of the EU/Government, and the original specification was created by fill the grant forms to fulfill the grant requirements. There were special forms for specification and after that it didn’nt seemed to be practical to use these standards. The size of this paper also constrains this content.
comments 13.
- Who were the stakeholders involved in defining the requirements?
response 13.
The development team consists of veterinarians and university agricultural specialists, as well as agricultural experts from the test site. These stakeholders were involved in the preparation of the specification.
comments 14.
- How were the scenarios defined?
response 14.
The scenarios were defined by the expert team, where the veterinarian and agricultural experts collected their proposals and the whole team, including the engineering experts defined the real development goals and scenarios.
comments 15.
- How were the experimental units selected?
response 15.
The experimental units were the partners in the consortium. They undertook to help test the devices by providing the location and the animals.
comments 16.
- How was the field testing planned and specified?
response 16.
We report on the detailed test results in a separate publication. This publication focuses on the engineering parts and there is not enough space to describe these details.
comments 17.
- How was the solution validated?
response 17
A new paragraph was inserted:
“The sensor validation involved comparing the measured data to the original physiological characteristics whenever possible. The validation was primarily carried out using videos recorded by a camera system and HR measurements taken with Polar HR sensors.”
comments 18.
- Can the proposed solution be compared with similar solutions?
response 18.
Since there was no significant quantity of the other commercial sensors in our collection, we were unable to conduct an experimental comparison.
A new paragraph was inserted:
“Based on the literature comparison, it is striving to be at the forefront of sensors in the market - if it is marketed as a product. It is capable of measuring all the characteristics defined in Table 5. in (Lee 2021), and it can also detect rumination and HR levels”.
comments 19.
- L141-146: Lacks details.
response 19.
It doesn’t define what type of details are missing.
comments 20.
- L154: What are the dimensions of the barn?
response 20.
The dimensions of the barn added to the article:
“The base area of each barn is approx. 110 m x 22 m.”
comments 21.
SYSTEM DESIGN
- Except for the communication aspects, section 2.1 lacks references.
response 21
6 references were inserted.
comments 22.
- There is no mention of approval from an Animal Ethics Committee.
response 22.
There was approval and later this have been inserted into the material.
comments 23.
Figure 1: "Algorhithm"
response 23.
Fixed.
comments 24.
Some acronyms were not introduced: LTE, POM-C, MEMS, LoRaWAN.
response 24.
Fixed
comments 25.
L157. You should define what is "necessitating strategic placement of gateways".
response 25
More details added: “In the experiment the gateway was installed to the center of the barn, to a height of approx. 5 m. According the Received Signal Strength Indication (RSSI) values of the data packets this resulted reliable connection. For long-term use with more barns the necessity of new gateways can be determined based on the RSSI of packets from boluses of cattle being in other barns.”
comments 26.
L169. There is little discussion on the energy efficiency trade-offs between non-rechargeable and rechargeable batteries or alternative power sources such as energy harvesting, which has gained attention in remote sensor applications.
response 26
More details were added: “Using rechargeable batteries was also considered, but rejected. First, recharging is problematic. Wireless energy transfer exists, but impractical for this application. Energy harvesting devices would either produce inefficient amount of power (e.g. thermoelectric or RF radiation based), or make the final design very error prone (e.g. mechanical ones).
Second, lifetime of rechargeable batteries are much less than non-rechargeable ones, especially when they are charged and discharged with very low currents and kept almost fully charged. Non-rechargeable batteries, on the other hand, can serve for extended time, when discharged with small currents.”
comments 27.
L175. The discussion could be enhanced by providing data or references to support the material's performance in similar applications. Additionally, the long-term degradation of the material due to digestive fluids or extreme environmental conditions should be addressed. You should include references to studies that validate the use of POM-C or similar materials in comparable biological environments. Discuss the results of any mechanical stress tests or long-term durability studies to assure readers of the material's suitability.
response 27.
One reference is inserted and an additional paragraph.
The POM-C material is officially tested in wet environments and it has official certificate for usage in the food industry. The rumen’s conditions are wet and only weakly acidic >4.0 pH and in this cases it is already tested. It is not the object of this research. The object of tis research is the stress-tests of the whole device in weakly acidic solution at 30-45°C temperature and shaking mechanical treatments. The tests were successful in one year duration.
comments 29.
L194. The text lacks references for field validation data or simulation results to support the claimed communication performance. While the use of LoRaWAN is practical, it would be valuable to compare it with other low-power communication alternatives like NB-IoT or Sigfox, especially in the context of coverage, scalability, and power consumption.
response 29
Comparison of different low-power wireless data communication protocols was done in a different study, connecting to our work. In its full length, it is beyond the limitations of this paper.
comments 30.
L198. The battery life estimation is well-detailed. However, it would be more robust if supported by real-world usage scenarios or relevant references.
response 30
Ongoing tests are running as a part of the installed bolus system. Results for the exact, real-world lifetime of the battery however require more time.
comments 31.
How will the device be monitored for battery degradation or sensor malfunctions?
response 31
“The system continuously checks the battery's charge and sends this information hourly to the server. Sensor malfunctions are not systematically checked, but the missing measurements are detectable on the server side.”
comments 32.
RESULTS
L227- You should explain more explicitly why each component (e.g., SoC, radio chip, MEMS sensor) was chosen. Include details about trade-offs (e.g., power consumption, cost, signal penetration) for a more rigorous discussion.
response 32
Selection of the actual electronic components was the result of a previous planning process. Many factors of the components were considered, e.g. accuracy, energy consumption, price and availability. The detailed selection process was included in a different study and is beyond the scope of this article.
comments 33.
L258. The results of validation are not presented. You should include quantitative validation of the system, such as comparing the accuracy of bolus data with the external sensor data. Provide error rates or statistical measures (e.g., correlation, precision, recall) to strengthen the results.
Discuss any anomalies or limitations observed during testing and how these might affect real-world deployment.
response 33.
The purpose of the experimental bolus sensor was algorithm development. This tool provided the native data needed for the development of programs running on the microcontroller. The only observed limitation was that during the preliminary experiments, there was no detection of a heart rate value above 150 bpm.
comments 34.
L246. You should include a comparison of theoretical battery life with actual performance from field experiments.
response 34
Ongoing tests are running as a part of the installed bolus system. Results for the exact, real-world lifetime of the battery however require more time. See comment 31 for additional details.
comments 35.
L261. You should provide a detailed description of the preprocessing algorithms used (e.g., which techniques or models were applied and why). Include results on how effective these algorithms are in reducing data volume while maintaining accuracy.
response 35.
Line 261 points to the discussion of the experimental bolus sensor, which did not do any preprocessing, but sent raw, unprocessed data. This data was helpful during algorithm development. The developed algorithms, which are running on the actual bolus, are discussed from subsections from 3.2.2. to 3.2.7.
comments 36.
L274. While the system records data hourly, there is no discussion on how often this data needs to be transmitted for actionable insights. You should provide a trade-off analysis between data transmission frequency and system responsiveness. Explain how hourly transmissions impact real-time alerting and decision-making.
Address data loss scenarios and how the system can mitigate these (e.g., through data buffering, error correction, or redundancy).
response 36.
The experimental bolus record data continuously and transmit them immediately, without any compression.
comments 37.
Include metrics for evaluating algorithm performance, such as data compression ratios or computational load on the microcontroller.
response 37.
The following text for the compression ratio calculation is added:
“The preprocessing algorithms can be considered as a special kind of (lossy) data compression. With a fixed measurement setup the compression ratio can be calculated as follows.
\[r_c = \frac{d_o}{d_i},\]
\noindent where $r_c$ is the computed compression ratio, $d_o$ is the number of output bytes and $d_i$ is the number of input bytes, during an arbitrary amount of time. For the calculations a one hour period will be used. In this case $d_o = 51$ bytes is the size of a single radio packet and $d_i$ is the amount of data measured by the sensors during this period. With 12.5 Hz sample rate, 3 axes and 2 bytes per axis the accelerometer generates $3600 \cdot 12.5 \cdot 3 \cdot 2 \cdot = 270000$ bytes. The temperature sensor does a measurement every 10 minutes (6 times each hour) and it generates 2 bytes each time, which means 12 bytes per hour. As a result, $d_i = 270012$ bytes and $r_c \approx 1.9 \cdot 10^{-4}$.”
comments 38.
Provide a comparison of the proposed system with other commercial or experimental bolus sensors. Highlight the specific advancements this system brings, such as improved data accuracy, longer lifespan, or new data insights.
response 38
The properties and capabilities of the proposed bolus system are discussed in the article in detail. The exact, measured differences between the proposed system and other commercial products or experiments are beyond the scope of this article.
comments 39.
Further elaborate on the importance of heart rate monitoring and its practical applications in livestock management. Include references to studies that emphasize the role of heart rate in detecting specific health conditions (e.g., stress, illness).
response 39.
A new paragraph was inserted.
"The connection between HR and stress has been shown in several studies, making HR detection an important feature. Thus, stress can be detected through the HR, which affects the animal's reproductive capacity and milk production. The HR can also be helpful in monitoring the state of health and calving."
Reviewer 2 Report
Comments and Suggestions for Authors
The article is interesting and provides suggestions for improvements in the variables to be measured by Bolus sensors and also in terms of data transfer. However, little information is provided regarding the issue of reducing the size or ease of application of this type of sensor. I also missed an economic evaluation of the bolus presented, with the possible costs of the components and sensors chosen. I believe that adding more information on these issues would make the article more complete.
Author Response
We are grateful for your thorough and forward-looking review of our article, and for your valuable comments and questions. We hope that our responses, additions, and revisions to the article will be accepted.
comments 1
The article is interesting and provides suggestions for improvements in the variables to be measured by Bolus sensors and also in terms of data transfer. However, little information is provided regarding the issue of reducing the size or ease of application of this type of sensor. I also missed an economic evaluation of the bolus presented, with the possible costs of the components and sensors chosen. I believe that adding more information on these issues would make the article more complete.
response 1 The requested information has been placed in the publication.
“The lower size limit of the bolus is the experimental size at which the device no longer remains in the rumen, but passes through the digestive system or returns to the esophagus during rumination, so it is not worth reducing the size of the device too much. In practice, it is worth choosing the size of the device in such a way that it is compatible with one of the already tested and marketed applicators and can be easily administered with it.”
“The device does not contain expensive components, so the cost of the prepared sensor is comparable to sensors already on the market, for example, a neck transponder.”
Reviewer 3 Report
Comments and Suggestions for Authors
The main question addressed by the research is the development of a versatile rumen bolus sensor for dairy cattle, focusing on conceptual design, sensor modalities, mechanical design, power supply, communication solutions, and sensor detection algorithms. The research aims to improve continuous and complex monitoring in precision livestock farming by evaluating the opportunities and trends in this technology​.
The original and relevant parts of the research focus on the innovative use of rumen bolus sensors with multi-modality functionalities, such as heart rate detection, motion, and rumination algorithms, which are largely unexplored for long-term cattle monitoring. The study addresses gaps in integrating multiple sensor technologies within a single device that can operate for the animal's entire lifespan, unlike external sensors which degrade over time.
Introduction:
- Line 25 to 27 – No one of those references tested the discomfort of the Bolus inside the cows. The phrase “The advantage of these sensors is that they can stay in the animal for a long 25 time, they don’t get lost or fall off, and the animal won’t feel any discomfort when wearing 26 them [8] [19].” Isn’t that true, how do you know they won’t feel any discomfort?
- The introduction could emphasize more about the comfort of bovines using bolus an invasive sensor.
Materials and Methods:
- Please provide more details about the environmental conditions (barn design, temperature, and humidity) under which the sensors were tested.
- Since the bolus is meant for long-term monitoring, the study could benefit from explicitly stating how long-term tests were conducted over different life stages of the animals (growth or reproductive phases)
Results are well described but need to be more discussed with other references.
Author Response
We are grateful for your thorough and forward-looking review of our article, and for your valuable comments and questions. We hope that our responses, additions, and revisions to the article will be accepted.
comments 1.- Line 25 to 27 – No one of those references tested the discomfort of the Bolus inside the cows. The phrase “The advantage of these sensors is that they can stay in the animal for a long time, they don’t get lost or fall off, and the animal won’t feel any discomfort when wearing them [8] [19].” Isn’t that true, how do you know they won’t feel any discomfort?
 response 1 The bolus technology is relatively old and at the beginning phase it was evaluated in details. An additional reference was inserted into the text, which refers the research of the bolus technology on relevant number of animals. It investigates the retention of the bolus depending on the size, and specific gravity of the bolus, the age and gender of the animal. We haven’t got direct information of the possible discomfort, but both based on theoretical considerations, both by measurement it is rightly assumed. During the experiments there was no any detected difference between bolus containing animals and not containing animals in physiological parameters and behavior.
comments 2- The introduction could emphasize more about the comfort of bovines using bolus an invasive sensor.
response 2. A short description is inserted, more details with the bolus technology. It is very important that bolus technology is not invasive, because without any hurt it is applicable, the cow can simply swallow it.
comments 3- Please provide more details about the environmental conditions (barn design, temperature, and humidity) under which the sensors were tested.
response 3. The primary goal of this publication is to describe the technical aspects of the bolus development. The testing is a very large and complex process which could be a content of several other publications. Because this conceptual consideration it was only roughly described the testing, essentially just strengthened by the fact of the tests. In this paper, a short description was added to the testing conditions.
comments 4- Since the bolus is meant for long-term monitoring, the study could benefit from explicitly stating how long-term tests were conducted over different life stages of the animals (growth or reproductive phases)
response 4. The primary goal of this publication is to describe the technical aspects of the bolus development. The testing is a very large and complex process which could be a content of several other publications. Because this conceptual consideration it was only roughly described the testing, essentially just strengthened by the fact of the tests. In this paper, a short description was added to the testing details, the age of animals.
Because this conceptual consideration it was only roughly described the testing, essentially just strengthened by the fact of the tests.
As the main focus of this paper to describe the technical aspects of the bolus development, and the testing is a very large and complex process which could be transcends this publication, it is only roughly described in the results chapter. The tests were conducted continuously since the August of 2022. The test were conducted on adult animals, included their reproductive periods. The animals participated in the experiments were kept with the control animals in the same barn with natural conditions. The temperature and humidity was continuously measured during the experiments.
comments 5. Results are well described but need to be more discussed with other references.
response 5. New references were added to the publication.
Round 2
Reviewer 1 Report
Comments and Suggestions for Authors
The authors properly addressed my comments from the first round of review.
There are just a few minor revisions:
L14. Do you have some restriction about using industry 5.0?
L78. AI acronym must be defined.
L98, You should replace "chapter" by "section".
L104. HR acronym must be defined.
L216. RF acronym must be defined.
L217. You should replace "less" by "shorter".
L506. MLP acronym must be defined.
Author Response
Thank you for your help in improving our article. We have made the suggested changes.